# Contextual Interaction for Argument Post Quality Assessment

**Yiran Wang**[1,2]**, Xuanang Cheng**[1,2]**, Ben He**[*1,2]**, Le Sun**[*2]
[1]University of Chinese Academy of Sciences, Beijing, China
[2]Institute of Software, Chinese Academy of Sciences, Beijing, China
{wangyiran20, chenxuanang19}@mails.ucas.ac.cn,
benhe@ucas.ac.cn, sunle@iscas.ac.cn

## Abstract

Recently, there has been an increased emphasis on assessing the quality of natural language arguments. Existing approaches primarily focus on evaluating the quality of individual argument posts. However, they often fall short when it comes to effectively distinguishing arguments that possess a narrow quality margin. To address this limitation, this paper delves into two alternative methods for modeling the relative quality of different arguments: 1) Supervised contrastive learning that captures the intricate interactions between arguments. By incorporating this approach, we aim to enhance the assessment of argument quality by effectively distinguishing between arguments with subtle differences. 2) Large language models (LLMs) with in-context examples that harness the power of LLMs and enrich them with in-context demonstration. Through extensive evaluation and analysis on the publicly available IBM-Rank-30k dataset, we demonstrate the superiority of our contrastive interaction approach over state-of-the-art baselines. On the other hand, while LLMs with in-context examples demonstrate a commendable ability to identify high-quality argument posts, they exhibit relatively limited effectiveness in quantifying argument quality and distinguishing between arguments with a narrow quality gap. Code is available at https://github.com/ucasYW/Contextual-Interaction-for-AQA.

## 1 Introduction

Given the highly subjective nature of argumentation, arriving at a standard answer for a contentious topic is often challenging, as diverse opinions exist. Consequently, assessing the quality of arguments is a complex task, as it necessitates assessing not only the relationship between arguments and the topic at hand, but also the quality of the argument itself (Wachsmuth et al., 2017b). Nowadays, there

are many debate websites on the Internet, such as Quora, Kialo, and Zhihu. Usually, those websites present multiple arguments in a masonry layout. People tend to view those arguments in a very short period of time, then give those arguments an overall estimation of their strength based on their impression of the relative quality of different arguments. In this regard, on the internet, when confronted with a large volume of text, people's evaluations of the quality of arguments often stem from quick impressions rather than careful consideration after thorough reading and thinking.

Argumentation is frequently perceived as a tool for facilitating various forms of reasoning, including decision-making and persuasion. However, these approaches often assume that the individuals involved will exhibit purely rational behavior. In contrast, human behavior is known to blend rational and emotional elements in guiding their actions. It has been suggested that a substantial link exists between the process of argumentation and the emotions experienced by the participants in that process. According to the study of social science (Benlamine et al., 2015; D'Errico et al., 2018; Li and Xiao, 2020; Hilton, 2008), the strength of an argument relies much more on the argument's emotional appeal rather than the argument's logical coherence, which means it is hard to capture the quality of arguments only with their own context. The first impression made by comparing different arguments plays an important role in assessing the argument's quality. However, how to model this comparing procedure remains uncharted. Indeed, most current AQ assessment approaches (Marro et al., 2022; Gurcke et al., 2021a; Wachsmuth et al., 2016; Persing and Ng, 2017) consider an argument's quality based on its own context under the restricted perspective of logical, rhetorical, or dialectical. Only a few works tried to solve the

AQ assessment problem through a monolithic view (Fromm et al., 2022). Some tried to capture a feature that may affect the quality of an argument, such as argument structure (Li et al., 2020) and discourse structure (Liu et al., 2021). However, revealing those features requires extra annotation costs under the guidance of linguistic specialists which makes those methods hard to be applied to large-scale datasets. Since argument quality is a relative concept, it's hard to distinguish the slight difference between them, especially for arguments with subtle differences.

This study presents a novel investigation into measuring the nuanced differences between arguments for quality assessment. Specifically, we examine the contextual interaction of content between various argument posts to enhance the quality assessment process. We explore two different approaches to simulate the comparison of different contexts as humans evaluate multiple arguments: 1) Supervised contrastive learning for cross-argument interaction that pulls together those arguments with similar quality, while pushing apart other arguments whose quality is at both ends. 2) Utilization of Large Language Models (LLMs) with in-context examples for the AQ assessment. Extensive evaluation on the standard IBM-Rank-30K dataset (Gretz et al., 2020) demonstrates that contrastive learning surpasses state-of-the-art baselines in terms of overall assessment accuracy across arguments of varying quality ranges, as well as in distinguishing arguments with similar quality. While LLMs with in-context examples exhibit effectiveness in recognizing arguments at extreme quality ends, they fall short compared to contrastive learning when quantifying argument quality and differentiating between arguments with close quality gaps.

Major contributions of this paper are: 1) An investigation of modeling contextual interaction for the AQ assessment. 2) Proposal of a contrastive interaction approach that outperforms state-of-the-art baselines on the IBM-Rank-30k dataset. 3) An exploration of LLMs with in-context examples for the AQ assessment underscores the limited performance of LLMs in quantitatively recognizing the subtle difference between arguments.

## 2 Related Work

**Argument Quality (AQ) Assessment**. The problem of creating a convincing argument has its origins in ancient Greece, where the persuasiveness of arguments was discussed through dialectic and rhetoric (Aristotle and Kennedy, 2006). Based on classical theories on arguments (Johnson and Blair, 2006; Hamblin, 1970; Perelman and Olbrechts-Tyteca, 1969; Eemeren and Grootendorst, 2003), Wachsmuth et al. (2017b) identify logical, rhetorical, and dialectical as the three aspects of AQ. With recent progress in natural language processing, AQ has been studied and applied in various domains, including student essays (Wachsmuth et al., 2016), news editorials (El Baff et al., 2020), and social media discussions (Wachsmuth et al., 2017c; Skitalinskaya et al., 2021).

In the current research on AQ assessment, the main focus has been on studying it as a sub-task in Argument Mining (AM). However, due to the extreme subjectivity of AQ, there is no clear definition for it. As a result, it is believed that there are various factors that can influence AQ. Wachsmuth et al. (2017b) summarized 15 factors that affect argument quality, categorizing them into logical, rhetorical, and dialectical aspects. Gurcke et al. (2021b) assessed argument quality specifically in terms of sufficiency with human efforts, hypothesizing that the conclusion of a sufficient argument can be derived from its premises. Li et al. (2020) assessed the persuasiveness of arguments by analyzing their argument structure using a factor graph model. On the other hand, Singh et al. (2021) focused on explicating implicit reasoning (warrants) in arguments with the help of trained experts. Falk and Lapesa (2023) attempted to enhance the AQ assessment performance by incorporating knowledge from other dimensions into the prediction process through multi-task learning. These studies consider each perspective of argument quality separately, which limits their holistic view of the concept. Additionally, some of these approaches require additional annotations (Marro et al., 2022), bringing difficulty in application to large-scale datasets.

**Cross-document Interaction**. The idea of modeling cross-document interactions has been widely explored in machine learning and learning to rank tasks. Pang et al. (2019) proposed SetRank, which utilizes a self-attention mechanism to capture local context information from cross-document interactions and learn permutation equivalent representations for the input documents. In a related area, van den Oord et al. (2018) introduced unsupervised contrastive learning, a method for extracting useful representations from high-dimensional data. This

technique has been influential in various domains and has shown promising results in learning meaningful representations. Building on the concept of contrastive learning, Khosla et al. (2020) extended it to the supervised setting, enabling effective utilization of label information.

**Large Language Models with In-context Examples**. In-context learning (ICL) (Honovich et al., 2022) has achieved tremendous success on large language models (LLMs). The main concept behind in-context learning is to leverage analogies for learning. In-context learning uses a small number of examples to create a demonstration context, often expressed through natural language templates (Min et al., 2022). Specifically, a query question and demonstration context are concatenated to form a prompt, which is then inputted into an LLM for prediction. Besides, to enhance the efficiency and performance of existing LLMs, especially for non-opensource APIs like ChatGPT and GPT-4, OverPrompt (Li et al., 2023) enables parallel processing of multiple inputs within a single prompt using in-context learning. Liu et al. (2022) aim to explore a more effective strategy for carefully selecting in-context examples, which can amplify GPT-3's in-context learning capabilities more efficiently than random sampling. Recently, some studies have employed LLMs for ranking and quality assessment tasks. Sun et al. (2023) examined the use of generative language models like ChatGPT and GPT-4 for relevance ranking in Information Retrieval (IR). Kocmi and Federmann (2023) utilized the GPT series models to assess the quality of machine translation and proposed the GEMBA metric.

## 3 Method

### 3.1 Overview

To imitate the reaction when humans read different arguments, we explore two alternative approaches for this comparing procedure, especially for the differentiation between arguments with a close quality gap. The first approach is based on supervised contrastive Learning (CL) (Khosla et al., 2020), which aims to maximize the similarity between similar pairs of samples, while simultaneously minimizing the similarity between dissimilar pairs. Additionally, a reasoning module based on graph neural networks is introduced to leverage the discourse relation and logical structure of arguments for their quality assessments.

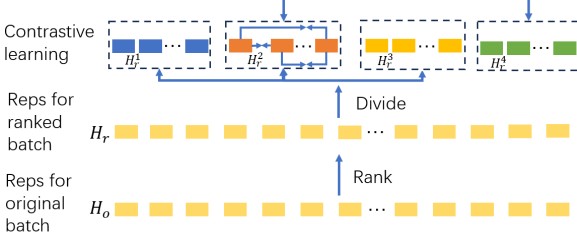

Figure 1: The supervised contrastive learning for cross-argument interaction. The representations (Reps) of arguments with median quality are pulled closer ($\rightarrow\leftarrow$), but pulled far ($\leftarrow\longrightarrow$) from other arguments whose quality are at both ends.

We explore an alternative approach that leverages the capabilities of LLMs. Specifically, we utilize LLMs with in-context examples, which can be viewed as a means of incorporating contextual interaction as prompted demonstrations. In this approach, the models analyze and process text while considering the provided examples within a given prompt. To imbue the LLMs with knowledge of argument quality, we supply the models with example arguments of varying quality levels in the prompts. We then prompt the model to assess and rate the provided arguments based on the standards set by those examples.

### 3.2 Supervised Contrastive Learning for Cross-argument Interaction

The architecture of our proposed supervised contrastive learning method is shown in Figure 1. During the training phase, each original batch $\mathcal{B}_o$ consists of $4n$ arguments from the training set, shuffled in random order. Each argument is concatenated with its corresponding topic using the delimiter [SEP] tokens, and a [CLS] token is added to the head of the sequence: *[CLS] Topic [SEP] Argument [SEP]*. The argument representation, which serves as the input to the contrastive objective, remains consistent with the backbone architecture, such as RoBERTa (Liu et al., 2019). After being encoded, we extract the last layer's hidden units of the [CLS] token as the representations for arguments in the original batch $\mathcal{B}_o$, obtaining a representation list $\mathcal{H}_o = [h_{1,o}, h_{2,o}, \cdots, h_{4n,o}]$.

To determine the anchor candidate for arguments with median quality within the batch, the original batch $\mathcal{B}_o$ is ranked based on the descending order of their true labels (human-annotated scores) to be an ordered one $\mathcal{B}_r$, with the ordered representation list $\mathcal{H}_r = [h_{1,r}, h_{2,r}, \cdots, h_{4n,r}]$, simplified

as $[h_1, h_2, \cdots, h_{4n}]$. Then, we divide this ranked batch into four equal-size mini-batches, denoted as $\mathcal{B}_r^s$ and $\mathcal{H}_r^s$ in which $s \in [1, 2, 3, 4]$. For example, the representations in the second mini-batch is $\mathcal{H}_r^2 = [h_{2n+1}, h_{2n+2}, \cdots, h_{3n}]$. To increase the distinction between median arguments with two extreme ends, the second and third mini-batches are chosen as the anchor candidate sets, and the other two mini-batches only serve as negative sample sets. That is, we hope that the arguments in the second (or third) mini-batches could be as close as possible, and be as far away as possible from the arguments in the remaining three mini-batches. The contrastive loss is defined as follows:

$$\mathcal{L}_{i,j}^s = -log \frac{e^{sim(h_i, h_j)/\tau}}{\sum_{h_k \in \mathcal{H}_r/\mathcal{H}_r^s} e^{sim(h_i, h_k)/\tau}} \quad (1)$$

$$\mathcal{L}_{cl}^s = \frac{1}{n} \sum_{h_i \in \mathcal{H}_r^s, h_j \in \mathcal{H}_r^s, i \neq j} \mathcal{L}_{i,j}^s \quad (2)$$

wherein $s$ is 2 or 3, $sim$ is the cosine similarity scores, and $\tau$ is the temperature parameter.

Apart from the contrastive alignment, the model is also trained under the ground-truth scores using a mean squared error (MSE) loss:

$$\mathcal{L}_{mse} = \text{MSE}(y_{pred}, y_{true}) \quad (3)$$

wherein $y_{pred}$ is the predictions of the model, and $y_{true}$ is the ground-truth scores.

The final loss is given by a combination of the contrastive loss and the MSE loss. The contrastive loss is to encourage the model to learn discriminative representations for anchor candidates and in-batch positives, while also promoting separation from the in-batch negatives. The MSE term measures the discrepancy between the model's predictions and the ground truth labels.

$$\mathcal{L} = \beta[\alpha \mathcal{L}_{cl}^2 + (1 - \alpha)\mathcal{L}_{cl}^3] + (1 - \beta)\mathcal{L}_{mse} \quad (4)$$

wherein $\alpha$ and $\beta$ are the weighting factors. By adjusting the value of $\beta$, we can control the relative importance of the contrastive loss and the MSE loss in the overall training objective.

### 3.3 Reasoning Using Discourse Relation

Similar to previous studies (Toledo et al., 2019; Lauscher et al., 2020; Habernal and Gurevych, 2016; Wachsmuth et al., 2017a; El Baff et al., 2018), we recognize the vital role played by the context and logical structure of an argument in the

domain of argument quality assessment. Some studies have investigated the impact of discourse relations on the quality of argumentation (Durmus et al., 2019; Li et al., 2020). In the field of reading comprehension, Huang et al. (2021) use Discourse-Aware Graph Network (DAGN) to capture advanced discourse features that can effectively represent passages to solve logic QA tasks. Herein, we further adapt this method to the domain of quality evaluation for arguments, leveraging graph neural networks to learn the discourse structure of contexts.

Specifically, as shown by the example in Figure 2, DAGN treats discourse units as nodes and constructs a graph structure with certain conjunctions serving as edges to generate the graph representation. Differently, we supplement common conjunctions (in Appendix A.1) used in arguments as edges when constructing the graph structure. Finally, the original [CLS] token of the argument and the generated graph representation is contacted as the representation of the argument.

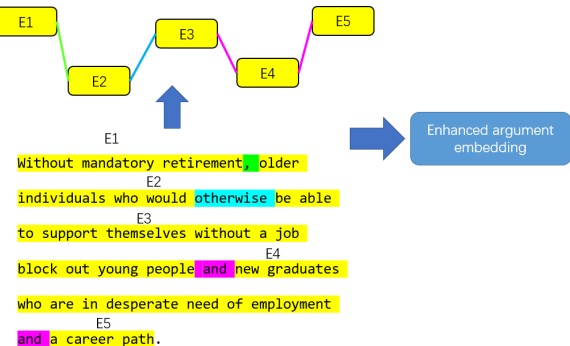

Figure 2: The example of the reasoning using discourse relation within the context of argument.

### 3.4 Large Language Models with In-context Examples

In this work, we employ two large language models, ChatGPT (gpt-3.5-turbo) and Davinci-003 (text-davinci-003). Both ChatGPT and Davinci-003 are based on the GPT-3.5 architecture, while Davinci-003 is an earlier version. Both of them offer excellent performance and can generate coherent and contextually relevant responses in many text generation tasks. Therefore, we aim to explore whether these two models exhibit equally outstanding performance in the AQ assessment task.

Our approach aims to emulate the comparative evaluation carried out by human readers when browsing debate forums or websites. To achieve

this, we adopt a method that incorporates knowledge of argument quality, designing the prompt with in-context examples. Examples within the prompt can be considered as posts that humans have already seen and their corresponding evaluations in their minds when they browse different arguments. Figure 3 shows the detailed prompt for AQ ranking and comparison task. For the prompt, we first provide example arguments along with their corresponding topic and score. Then, we provide the task instructions. Finally, we present the argument(s) to be evaluated or the arguments pair to be distinguished, followed by "Score:" or "Better argument: ", indicating that the model is required to complete the score or judgment.

## 4 Experiments

### 4.1 Experimental Setup

**Dataset**. IBM-Rank-30k (Gretz et al., 2020) is a large-scale argument dataset containing 30k arguments over 71 topics. Each argument in the dataset is annotated with a continuous quality score between 0 and 1, and we use weighted average quality scores as ground-truth labels since they negate the influence of non-reliable annotators. The training subset contains 49 topics with 20,974 arguments. The Dev subset contains 7 topics with 3,208 arguments and is used for tuning hyper-parameters and determining early stopping. The test subset contains 15 topics with 6,315 arguments.

**Task setup.** We evaluate the effectiveness of incorporating contextual interaction for argument quality (AQ) assessment on two tasks, namely, the AQ ranking, and the AQ comparison.

The **AQ ranking** task follows the official setup of IBM-Rank-30k. Given a list of arguments $A = [a_1, a_2, ..., a_n]$ and their corresponding topics $T = [t_1, t_2, ..., t_n]$, the task is to assign a ranking for those arguments based on the descending order of their quality score $S(a_i, t_i)$. The result of the argument quality ranking task is evaluated with Pearson correlation (Cohen et al., 2009), Spearman correlation (Wissler, 1905), and NDCG (Normalized Discounted Cumulative Gain) (Järvelin and Kekäläinen, 2002) on all test samples.

The **AQ comparison** task involves predicting the argument of higher quality in a given argument pair, denoted as $(A_1, A_2)$. To evaluate the performance of models on this task, we conduct experiments using three sets of argument pairs, each consisting of 2,000 pairs. These pairs were categorized based on the difference in argument quality scores. The categories include pairs with a score difference of less than 0.25, ranging from 0.25 to 0.5, and exceeding 0.5. The objective of this task is to assess how the model performs in terms of differentiating between argument posts in different levels of quality gaps. Accuracy, the percentage of correct predictions made by the model over all argument pairs, is used to evaluate this task.

**Implementation details.** We use RoBERTa-base (Liu et al., 2019) and BERT-base (Devlin et al., 2019) as our backbone models. The hyper-parameters $\alpha$ and $\beta$ in Eq.4 are set to 0.5 and 0.8, and the temperature in Eq.1 is set as 0.1. We fine-tuned the RoBERTa-base model for 5 epochs with a learning rate of 2e-5 and a batch size of 32, which align with the settings of the BERT model in (Gretz et al., 2020). The vanilla model and models with DAGN are trained by MSE loss, while models with CL are trained based on the loss in Eq.4.

As for demonstrations to LLMs, we employed two settings for example numbers $N$. The first approach includes three examples consisting of a high-quality argument (1 point), a medium-quality argument (0.5 points), and a low-quality argument (0.1 points). The second setting involves providing ten examples in the prompt, with scores ranging from 0.1 to 1. The smallest difference in scores between each pair of examples is approximately 0.1 points. We experiment with these two settings for both AQ ranking and comparison tasks.

In AQ ranking tasks, we instruct LLMs to give a quality score of the input argument, and we test two settings on candidate arguments numbers $S$: 1) $S = 1$, rating individual argument, and 2) $S = 10$, rating groups of ten arguments together, to evaluate the impact of the candidate arguments number on AQ ranking performance. In AQ comparison task, to recognize if the LLMs have the ability to distinguish the quality of arguments, we ask the LLMs to select the argument with better quality between the two options, that is, we instruct them in the prompts to return the number (1 or 2) corresponding to the higher-quality argument according to the specified format. The prompts used can be found in Figure 3. Due to the randomness of scoring arguments using LLMs, we conduct five experiments separately and report the average score or majority vote of the five sets of experiments as the final results. In the naming of model variants, $k$E and $k$S stands

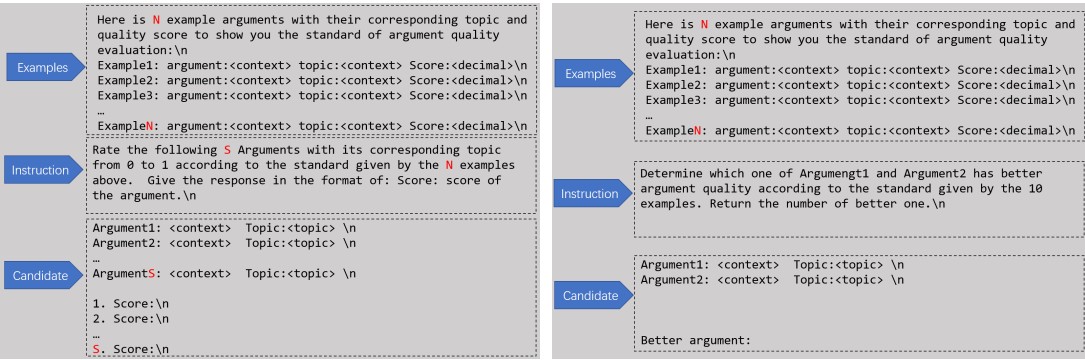

(a) Prompt for AQ assessment  (b) Prompt for AQ comparison

Figure 3: The prompt for AQ assessment and comparison, where *N* represents the number of examples, and *S* represents the number of arguments to be evaluated.

for the example arguments presented to the LLM, and the arguments that the LLM is asked to score for once. For instance, ChatGPT-3E-10S stands for ChatGPT which is prompted to score a batch of 10 arguments while being shown with 3 examples.

**Comparison models.** We employ the following baselines in our evaluation:

**SVM BOW** is a support vector regression ranker (Gretz et al., 2020). The training set is composed of the most frequent 1000 tokens and utilizes an RBF kernel and bag-of-words features.

**Arg Length** (Gretz et al., 2020) evaluates argument quality based on its length in characters, following the intuition that longer texts may provide more detailed explanations.

**RoBERTa and BERT** (Favreau et al., 2022) concatenate the argument with its corresponding topic, and generate a quality score.

**TFR-BERT** with ensemble losses achieves state-of-the-art effectiveness on IBM-Rank-30k as shown in (Favreau et al., 2022). This approach is built upon the work by (Han et al., 2020), which incorporates several ranking losses in TFR-BERT. Favreau et al. (2022) applies a similar technique to evaluate argument quality by combining the output of multiple TFR-BERT models, each trained with a distinct ranking loss.

**RoBERTa w/ own adpt** (Falk and Lapesa, 2023) is a recent approach that integrates knowledge from different dimensions into the prediction process using multi-task learning. RoBERTa-base is used as the backbone model for all the dimensions.

**Dual BERT w/ spark (ZS)** (Deshpande et al., 2023) involves four types of augmentations for the AQ prediction by offering feedback, deducing hidden assumptions, providing a similar-quality ar-

gument, or presenting a counterargument. Dual BERT was used as the backbone model.

The outcomes obtained from SVM BOW and Arg Length are referenced from (Gretz et al., 2020). The results of BERT and TFR-BERT are directly cited from (Favreau et al., 2022). Additionally, RoBERTa w/ own adaptar and Dual BERT with Spark (ZS) are cited from (Falk and Lapesa, 2023; Deshpande et al., 2023), respectively.

### 4.2 Experimental Results

Tables 1 & 2 present the results obtained for the AQ ranking task and the AQ comparison task, respectively.

**Our proposed contrastive argument interaction outperforms state-of-the-art baselines.** The results in Table 1 show that RoBERTa's performance is superior to that of BERT. RoBERTa achieves a Pearson correlation of 0.5283 and a Spearman correlation of 0.4858. It also can be seen that our proposed contrastive learning (CL) for the argument interaction outperforms the baselines with a Pearson correlation of 0.5580 and Spearman correlation of 0.5186. According to NDCG@K, our methods outperform baselines by a significant margin. Moreover, RoBERTa + DAGN and RoBERTa + CL achieve a prominent performance with a Pearson correlation of 0.55 and Spearman correlation of 0.51. The prominent performances of these 2 models reveal that reasoning and comparing both have a large impact on argument quality. The experimental results indicate that CL can consistently improve the AQ ranking performance over vanilla RoBERTa and BERT models. For the AQ comparison task, we exclusively incorporate RoBERTa-based models, excluding LLMs, as they have demonstrated su-

| - | Model | Pearson | Spearman | NDCG@5 | NDCG@10 | NDCG@15 |
|---|---|---|---|---|---|---|
| Baselines | RoBERTa | 0.5283 | 0.4858 | 0.9304 | 0.9304 | 0.9427 |
| | BERT[†] | 0.5200 | 0.4800 | 0.8500 | 0.8700 | 0.8600 |
| | TFR-BERT Ensemble Losses[†] | 0.5200 | 0.4700 | 0.8900 | 0.8900 | 0.8800 |
| | RoBERTa w/ own adpt[†] | 0.4850 | - | - | - | - |
| | Dual BERT w/ spark (ZS)[†] | 0.2742 | 0.2854 | - | - | - |
| | SVM BOW[†] | 0.3200 | 0.3100 | - | - | - |
| | Arg-length[†] | 0.2100 | 0.2200 | - | - | - |
| BERT-based | BERT w/ CL | 0.5278 | 0.4879 | 0.9191 | 0.9163 | 0.8872 |
| | BERT w/ DAGN | 0.5292 | 0.4943 | 0.9222 | 0.9258 | 0.9264 |
| | BERT w/ DAGN and CL | 0.5375 | 0.4949 | 0.9330 | 0.9372 | 0.9388 |
| RoBERTa-based | RoBERTa w/ CL | 0.5580 | **0.5186** | 0.9740 | 0.9614 | 0.9634 |
| | RoBERTa w/ DAGN | 0.5511 | 0.5112 | 0.9654 | 0.9655 | 0.9645 |
| | RoBERTa w/ DAGN and CL | **0.5604** | 0.5174 | **0.9799** | **0.9769** | **0.9648** |
| LLMs | Davinci-003-3E-1S | 0.1013 | 0.1092 | 0.7758 | 0.7758 | 0.7758 |
| | ChatGPT-3E-1S | 0.1277 | 0.1382 | 0.7672 | 0.7672 | 0.7671 |
| | Davinci-003-10E-1S | 0.0411 | 0.0401 | 0.5901 | 0.5899 | 0.5899 |
| | ChatGPT-10E-1S | 0.1221 | 0.1509 | 0.6270 | 0.6270 | 0.6271 |
| | ChatGPT-20E-1S | 0.1130 | 0.1153 | 0.6657 | 0.6657 | 0.6658 |
| | ChatGPT-30E-1S | 0.0863 | 0.0713 | 0.6301 | 0.6303 | 0.6303 |
| | Davinci-003-3E-10S | 0.1680 | 0.1633 | 0.7781 | 0.7781 | 0.7780 |
| | ChatGPT-3E-10S | 0.1896 | 0.1903 | 0.8044 | 0.8044 | 0.8044 |

Table 1: Results of AQ ranking task. Our proposed models are based on contrastive learning (CL), discourse-aware graph network (DAGN) and large language models (LLMs). The best results are marked in bold, and the results of best baseline are underlined. † denotes that the results are directly cited from corresponding papers.

perior performance compared to other models. The result in Table 2 revealed that CL method achieves the highest accuracy in discerning argument pairs with a difference in scores within 0.25 and above 0.5, reaching 65.45% and 89.6%, respectively. This confirms that our proposed approach effectively enhances the model's ability to distinguish subtle differences and creates a clear distinction between arguments with intermediate scores and those at the extreme ends.

| Model | Accuracy | | |
|---|---|---|---|
| | D < 0.25 | 0.25 <D< 0.5 | D > 0.5 |
| ChatGPT-3E | 53.20% | 65.95% | 76.25% |
| ChatGPT-10E | 48.80% | 59.00% | 78.30% |
| ChatGPT-20E | 49.35% | 52.40% | 68.20% |
| ChatGPT-30E | 48.40% | 52.65% | 69.90% |
| Davinci-003-3E | 54.20% | 66.85% | 73.80% |
| RoBERTa | 63.15% | 75.85% | 88.35% |
| w/ CL | **65.45%** | 77.60% | **89.60%** |
| w/ DAGN | 64.45% | **78.35%** | 89.00% |
| w/ DAGN and CL | 64.75% | 78.25% | 89.45% |

Table 2: AQ comparison results. $D$ denotes the difference value of quality score between argument pairs.

**Performance of LLMs.** As shown by the results in Tables 1 and 2, despite the fact that LLMs have

demonstrated strong ability in text generation tasks (Bang et al., 2023; Shen et al., 2023; Laskar et al., 2023), they still have significant limitations compared to our models when it comes to quantifying text quality. Indeed, both Davinci-003 and ChatGPT underperform in the task of scoring text, whilst ChatGPT outperforms Davinci-003 as expected. After a closer look at the results, when there are 3 examples in a prompt, we find that Davinci-003 exhibits stronger randomness in predicting scores for low to medium scores and tends to give scores similar to those in the examples through observation of the predicted score distributions in the three experiments. On the other hand, ChatGPT provides more diverse predicted scores, indicating its stronger comprehension ability compared to Davinci-003. When we attempted to predict their quality by grouping candidate arguments into sets of 10, there was a slight improvement in the results compared to before. This indicates that the interaction between arguments is also effective in LLMs. When the number of examples in the prompt is increased to 10, the performance decreases compared to when there are only 3 examples, particularly evident in the NDCG metric. The performance deterioration becomes even more pro-

| Topic | Argument | Ground-Truth | RoBERTa | RoBERTa w/ CL |
|---|---|---|---|---|
| Factory farming | factory farming allows for greater food production to feed a growing population. | 0.95 | 0.97 | 0.93 |
| | Factory farming is a great way to get more food for everyone | 0.84 | 0.95 | 0.87 |
| Social media | social media allows access to social service resources, food pantries, various support groups, housing and many other viable resources. it helps maintain contact with family and friends. | 1 | 0.95 | 0.95 |
| | social media is a great tool to make business or personal connections | 0.86 | 0.94 | 0.88 |

Table 3: Examples on arguments with small difference, on which RoBERTa w/ CL can give more discriminative scores than vanilla RoBERTa.

nounced as the number of examples increases to 20 and 30. Upon observing the experimental results, one possible explanation is that due to the strong memorization capabilities of the GPT-3.5 model, it tends to heavily rely on the scores provided in the examples (details are shown in Appendix A.2). As the number of examples in the prompt increases, large language models tend to be more inclined to repeat the content of the examples from the prompt in their output, including the scores and text of the examples. This results in a failure of the final evaluation outcome.

Although the results in Table 2 indicate that the LLMs can distinguish between good and poor argument quality when there is a significant quality gap (with a difference in quality scores above 0.5), the accuracy of the LLMs drops to around 50% when the difference in argument quality is small (with a difference in quality scores below 0.25). This accuracy is close to that of random selection, which suggests that LLMs are more inclined to randomly select rather than make judgments based on understanding when the difference in argument quality is small. When the difference in argument quality is between 0.25 and 0.5, the comparison capability of the LLMs shows a significant improvement, reaching around 67%. However, there is still a 10% gap compared to the performance of the fine-tuned models. This phenomenon is also reflected in the performance of the LLMs on the correlation metrics for the quality ranking task, where it shows relatively poor results. However, it achieves a decent performance in terms of the NDCG metrics, which indicates LLMs are able to distinguish between good and poor arguments but struggle to differentiate small differences in quality.

## 5 Discussion

**Argument interaction helps distinguish the slight difference.** As a case study, two example

arguments in Table 3 express their opinions supporting factory farming. While both arguments discuss the advantages of factory farming in increasing agricultural output, slight differences in expression lead to variations in their quality scores. One argument receives a high-quality score of 0.95, while the other argument scores around 0.8, indicating slightly lower quality. When using RoBERTa-base without argument interaction, the model assigns a score of 0.97 to the argument with a score of 0.95. However, for the other argument that has similar content but slight flaws in expression, RoBERTa-base still assigns a high score of around 0.95. Meanwhile, after introducing contrastive learning as a means of argument interaction, the model maintains a score of 0.95 for the argument with a score of 0.95. Additionally, it can provide a score of 0.87 which is closer to the true score for the other argument. Similar results are observed for the topic of *Social media*. These two examples demonstrate how the argument interaction can enhance the model's ability to discern subtle differences in expressions.

**Distribution of predicted scores with argument interaction is closer to the real distribution.** The histogram presented in Figure 4 illustrates that the fine-tuned RoBERTa-base generates quality logits that are more polarized compared to the actual data distribution. In the ground truth, top arguments (score $\geq$ 0.9) make up about 34% of the distribution, while in the prediction without argument interaction, they account for nearly 50%. This suggests that the model fails to distinguish between decent and good arguments due to the difference in expression skills and the resulting emotional responses from readers. The histogram of the model with contrastive learning exhibits a distribution closer to the real data. It identifies top arguments at approximately 36%, indicating better differentiation between decent and good ar-

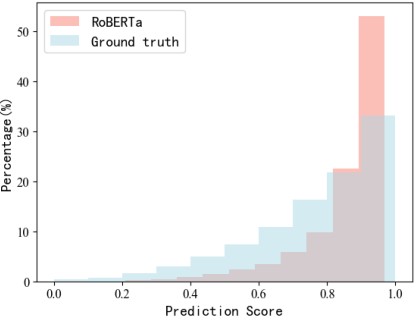 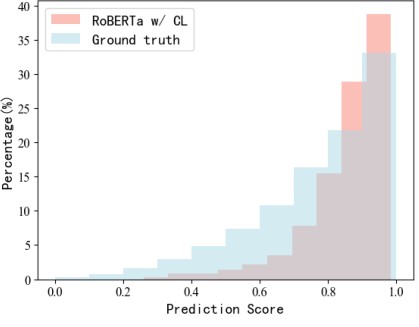

(a) The prediction results of RoBERTa  (b) The prediction results of RoBERTa w/ CL

Figure 4: Histogram of the prediction from RoBERTa and RoBERTa w/ CL compared with ground truth.

guments compared to the RoBERTa-base model. These findings demonstrate that incorporating contrastive learning enables the model to accurately capture subtle differences in quality and provide more precise assessments.

**The potential of LLMs in AQ assessment tasks.** LLMs have shown immense potential in argument quality (AQ) assessment tasks, particularly in argument comparison. As evident from the results in Table 2, when there is a significant difference (above 0.5) in the quality of arguments, large language models can achieve comparable resolution capability with only a few examples. However, despite this, there is a significant gap between large language models and fine-tuned models when the difference in quality between arguments is reduced. When the quality gap is small (below 0.25), the evaluation of quality by LLMs becomes closer to random guessing, indicating their limited ability to capture nuanced contexts by the LLMs involved in this study. Meanwhile, we have also observed that when it reaches a certain number of examples in the prompts (e.g., 20 examples), due to the excellent memorization ability of the large model, it tends to replicate text and scores that appear in the prompts in the answers. How to make the LLMs overcome the hallucination during the evaluation process, and return appropriate content, which can also help improve the performance of the large model in AQ tasks.

To evaluate the ability of LLMs to assess the quality of arguments under the *same topic*, we conducted experiments on the topic "We should adopt libertarianism". Due to the limited number of arguments available for a single topic (approximately 300 to 400), we conducted experiments on the argument quality comparison task by selecting 100 ar-

gument pairs for each of the three score differences under 3 examples. Despite the relatively limited experimental sample size, which led to some fluctuations in the results, the experimental findings are largely consistent with the conclusions obtained in Table 2.

This observation suggests that LLMs, despite their proficiency in open-ended text generation (Arora et al., 2022), may lack precision in assigning precise AQ scores, leading to relatively lower performance in AQ ranking tasks.

## 6 Conclusion

In this study, we explored two alternative methods for assessing the quality of natural language arguments: supervised contrastive learning and large language models (LLMs) with in-context examples. Comprehensive evaluation highlights the importance of considering contextual interactions for evaluating argument quality. We found that supervised contrastive learning, which captures intricate interactions between arguments, outperformed state-of-the-art baselines in evaluating argument quality. However, LLMs with in-context examples showed limitations in quantifying argument quality and distinguishing between arguments with a narrow quality gap.

## Acknowledgments

This work is supported by the National Natural Science Foundation of China under grant No. U1936207 & 62272439, and the Fundamental Research Funds for the Central Universities.

## Limitations

While the proposed contrastive learning approach enhances the effectiveness of AQ predictions, it is important to acknowledge that it may introduce additional computational costs. The inference cost increased as we introduced DAGN and CL. The phenomenon that LLMs failed in AQ ranking task can be attributed to several potential factors. Bias in training data, model interpretability, and the need for domain-specific fine-tuning are among the factors that require careful consideration and mitigation. We only conducted experiments on ChatGPT and Davinci-003, without testing GPT-4. From our experiments, it can be observed that LLMs hold great promise in AQ assessment tasks. While substantial prospects exist for LLMs in the domain of quality evaluation tasks, overcoming the aforementioned challenges remains an imperative subject for ongoing research.

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

# A Appendix

## A.1 Supplementary edges for graph construction

We have added five prepositions and conjunctions with logical meanings that occur in the corpus. They are "in case", "contrary to", "against", "so as to", and "in order to".

## A.2 Supplementary experiments conducted under different example scores

To demonstrate the susceptibility of large models to the influence of example scores and their tendency to concentrate predicted scores around the example scores in the context of argument text assessment, we conducted experiments on 1000 candidate arguments on ChatGPT and Davinci-003 with a prompt of three examples and one argument to be evaluated at each time. As shown in the Figure 5, the davinci-003 model was significantly affected by the example scores. As expected, chatGPT outperformed the davinci-003 model.

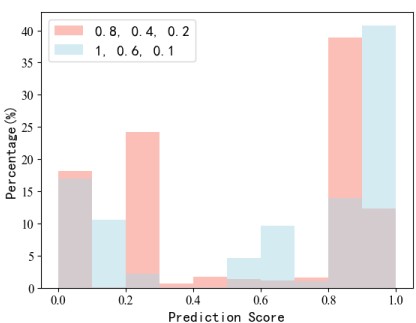

(a) Prediction distribution of Davinci-003

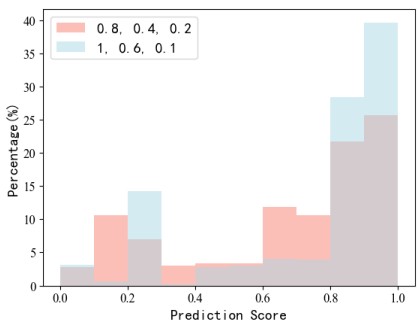

(b) Prediction distribution of ChatGPT

Figure 5: Prediction distribution of LLMs under two different example scores, namely, 0.8, 0.4, 0.2 and 1, 0.6, 0.1.