# OpenReview forum: "Contextual Interaction for Argument Post Quality Assessment"
_EMNLP/2023/Conference — EMNLP 2023 Main_

### Official Review · Reviewer_MHeS · 2023-08-04

**Soundness:** 4

**Excitement:**

4: Strong: This paper deepens the understanding of some phenomenon or lowers the barriers to an existing research direction.

**Missing References:**

none

**Paper Topic And Main Contributions:**

In this paper, the authors investigate the influence of contextual interaction on human assessment of argument quality. To this end, they propose to combine language models with supervised contrastive learning, as well as with a discourse-sensitive embedding of the argument. In addition, they investigate the potential of prompt-based approaches, where the context is part of the prompt. On the IBM-Rank-30k dataset, they evaluate the models’ ability for (1) argument quality ranking and (2) argument quality comparison. They show that the use of contrastive learning as well as discourse-sensitive argument embeddings with RoBERTa significantly outperforms previous approaches for (1), as well as strongly performs for (2), where it particularly excels in distinguishing arguments with only small differences in quality. They also find that prompt-based approaches perform significantly worse.

**Questions For The Authors:**

- A: CL and DAGN improve performance by a few percentage points. Therefore, the authors could discuss the limitations of incorporating contextual interactions on performance. What were the expectations in this regard? Was a larger increase expected based on the social
- B: Adding DAGN into the comparison in Section 5 could reveal further interesting insights.

**Reasons To Accept:**

- Evaluating argument quality, especially as a ranking task, and in cases where the quality between arguments does not differ much, is a hard task of computational argumentation. This work contributes an approach that outperforms previous SOTA results by incorporating contextual interaction.
- The paper is written in a very clear and well structured way.
- The approaches are well motivated and the experiments are comprehensible. The results are very clearly summarized and interpreted during the evaluation.
- The authors provide a very interesting analysis of the ability of LLMs (ChatGPT and Davinci) to distinguish arguments in terms of quality. In doing so, they reveal weaknesses of these highly praised approaches. Especially that the performance in distinguishing similar arguments is equivalent to random guessing is an exciting finding.
- The effect of contextual interaction is nicely illustrated in the paper using a case study and a comparison of the distribution of ground truth and predictions. This analysis supports the hypothesis of this paper.

**Reasons To Reject:**

- The paper's introduction appears to be its least strong aspect. In this section, the idea of contextualization could be emphasized more, along with better integration into social science research.
- The use of ChatGPT and Davinci is motivated by their good performance on text summarization. Here, it should be explained in more detail why the authors use this as an indicator for likewise good performance on AQ, since the relation to the task at hand is not straightforward.

**Reproducibility:**

5: Could easily reproduce the results.

**Reviewer Confidence:**

4: Quite sure. I tried to check the important points carefully. It's unlikely, though conceivable, that I missed something that should affect my ratings.

**Typos Grammar Style And Presentation Improvements:**

- 144: attempt
- 175: uses
- 186: aim
- 187: strategy
- Figure 1: augments → arguments?
- 257: the other
- 409: In the AQ

---

> ### Author Rebuttal · Authors · 2023-08-28
>
> We would like to sincerely thank you for your careful consideration of our paper and for providing valuable feedback. Please see below for our responses.
>
> R1. Strengthening the Introduction: We appreciate the reviewer's feedback regarding the introduction of our paper. We acknowledge the importance of emphasizing the integration of contextualization into social science research. In the revised version, we will highlight the significance of contextual interaction in computational argumentation and its relevance to social science research. For instance, how emotional expression in an argument affects people's first impression and subsequently influences their evaluation of the argument's quality. We appreciate any specific suggestions or insights to further enhance our work.
>
> R2. Justification of LLMs Choice: Considering the remarkable performance of LLMs in text summarization, which stems from their ability to understand textual content, it is reasonable to assume that this ability could also contribute to their effectiveness in accomplishing the AQA task. Based on this assumption, we posit that LLMs would exhibit favorable performance in AQA tasks.
>
> R3. Discussion of Performance Improvement Limitations: Our aim is to enable text interaction by introducing representations for structural analysis of text, emulating the comparative process humans undertake after reading and contemplating. Although this approach yielded the best performance, the improvements achieved were somewhat limited. We attribute this limitation to the concatenation of the [CLS] embedding with the DAGN embedding used to obtain representations of structure augmented argument. While this concatenation preserves semantic information and enhances structural details, it can introduce interference during contrastive learning, hindering the expected outcomes. From a sociological perspective, this limitation might be attributed to the greater significance placed on first impressions. Just as in our everyday lives, where the initial impression holds considerable weight, the subsequent impact of our thoughts may not be as influential as the role of that initial impression.
>
> R4. Comparison with DAGN: In our previous experiments, we attempted to introduce more words, including certain prepositions, as edges. The results indicated that finer decomposition of the structure does not necessarily lead to better performance. In fact, an excessive number of edges can result in a decline in performance.
>
> Typos & Presentation: Thanks very much for your thorough review. We will make the revisions to the paper based on your feedback.

---

### Official Review · Reviewer_eHo2 · 2023-08-05

**Typos Grammar Style And Presentation Improvements:** 1. Abstract and Intro, when read toge…
**Soundness:** 4

**Excitement:**

4: Strong: This paper deepens the understanding of some phenomenon or lowers the barriers to an existing research direction.

**Paper Topic And Main Contributions:**

The authors propose two methods for modelling argument quality: SCL and LLMs. Results provided show SOTA improvement by using CL for IBM 30K dataset, while LLMs do well in certain instances (when two arguments are different enough). Sufficient baselines are used for comparison.

**Questions For The Authors:**

1. Please refer to point 2 above, would love to hear thoughts on it!
2. In line 419, why is average a good metric?

**Reasons To Accept:**

1. The idea of comparing LLMs to CL in this specific domain is very innovative.
2. Very thorough testing and results that prove differences between the two methods- conclusions like LLMs guessing randomly when argument nuance is limited is derived through careful testing, and is an important conclusion that highlights the limitations of LLM, especially now, when LLMs and ChatGPT are in the spotlight.
3. Table 1 summarizes everything in a nice manner.
4. Table 3 and its finding is extremely interesting.

**Reasons To Reject:**

1. I think SCLs are explained fairly well, and achieve SOTA results. However, LLMs are not explored futher. Why LLMs produce random results when argument score difference is small is not explored (just an observation, I wouldn't reject it based on this).
2. Some details about LLMs can be useful- what examples are provided? How do you ensure they are truly representative? What topics are covered in one set? If not, did you try providing it different arguments for the same topic and then seeing if its still random?

**Reproducibility:**

4: Could mostly reproduce the results, but there may be some variation because of sample variance or minor variations in their interpretation of the protocol or method.

**Reviewer Confidence:**

3: Pretty sure, but there's a chance I missed something. Although I have a good feel for this area in general, I did not carefully check the paper's details, e.g., the math, experimental design, or novelty.

---

> ### Author Rebuttal · Authors · 2023-08-28
>
> We would like to sincerely thank you for your careful consideration of our paper and for providing valuable feedback. Please see below for our responses.
>
> R1. Exploring LLMs further: We acknowledge this concern and realize that a deeper exploration of LLMs and their tendencies is important. While the reason why LLMs producing random results when the argument score difference is small is not elaborated upon in the paper, our guess is that the LLMs used were trained on highly diverse corpus, and were perhaps not specialized for the AQA task, such that they lack the ability to capture subtle differences between arguments. Additionally, as observed in Appendix 3, the LLMs appear to have the tendency to memorize scores from examples rather than evaluating quality based on comprehension. This is also a contributing factor to the limitations of the large model's performance on AQ tasks. However, it can be possible to improve the LLMs' AQA ability by supervised fine-tuning on sufficient high quality training data.
>
> R2&Q1. Context details and data sources: We apologize for any ambiguity in our paper. The <context> arguments used as examples in our prompt are sourced from the IBM30k training set. As we described in section 4, Implementation details, we choose examples with diverse scores to enable them to represent arguments of varying quality.  As per reply to R2 of Reviewer gESR, we present arguments from different topics following prior works, as presenting arguments from the same topic does not appear to significantly change the results.
>
> Q2: We sincerely apologize for the confusion. To clarify, for the ranking task using LLMs, we utilize an approach where the average of three runs is used. As for pairwise quality comparison task, we employ a majority vote method. Due to the inherent randomness of large models, we employ a methodology of conducting 3 experimental runs and averaging the results. This improves the stability and robustness of the experimental outcomes.
>
> Typos & Presentation: Thanks very much for the suggestions. We will update our paper based on your feedback.

---

### Official Review · Reviewer_gESR · 2023-08-05

**Soundness:** 4

**Excitement:**

4: Strong: This paper deepens the understanding of some phenomenon or lowers the barriers to an existing research direction.

**Paper Topic And Main Contributions:**

The main contributions of this paper that stood out to me are as follows:
1. The idea that interactions between arguments can be used to gauge quality is a new one to me. The similarity between argument composition leading to an agreement of scores is an interesting direction to explore.
2. The approach of combining Supervised Contrastive Learning with DAGN is an innovative one.
3. The application of LLMs to the task of scoring arguments having been given examples of argument-score pairings is interesting.

**Questions For The Authors:**

A. Would be interested in a detailed description of the context provided to the LLMs to score arguments.
B. Do the topics that the arguments relate to have an impact in terms of scoring and if so, how is that captured in the model proposed?

**Reasons To Accept:**

The ideas presented in this paper are on a whole, innovative and logical.
1. Analyzing and understanding the structural similarity between arguments is quite an interesting concept, and the results shown in table 2 do suggest that the interactions that the author/s are trying to capture impact the scoring of said arguments.
2. While there are some issues with the approach taken by the author/s to utilize LLMs for argument quality detection (detailed below), I believe the approach as a whole is an interesting one.
3. The conclusion that LLMs can identify high quality arguments but cannot distinguish between similarly scored arguments well aligns with the capabilities of LLMs as of now.
4. Reasonable baselines are used and the SCL method shows improvement on SOTA.

**Reasons To Reject:**

There are some specific issues with the task of argument quality detection using LLMs -
1. The author/s take the average of three runs for this task. Considering the variance inherent in LLMs and the relatively small size of the context, it may be possible that three runs are not enough. It would be good to see the impact of more runs and whether that causes a change in the results reported.
2. More details need to be provided with regards to the context provided for the LLMs to be able to assign a quality score to an argument - are they arguments from IBM30k? If so, are they from the same topic or from different ones and does that impact the score assigned?
3. The author/s provide a maximum of 10 arguments in the context for LLMs to score arguments. This seems a bit random, especially considering that IBM-30K has nearly 30,000 scored arguments. It may be interesting to see if the scoring is improved on provision of more context and whether there is a drop off in terms of improvement after a certain threshold.

**Reproducibility:**

4: Could mostly reproduce the results, but there may be some variation because of sample variance or minor variations in their interpretation of the protocol or method.

**Reviewer Confidence:**

3: Pretty sure, but there's a chance I missed something. Although I have a good feel for this area in general, I did not carefully check the paper's details, e.g., the math, experimental design, or novelty.

---

> ### Author Rebuttal · Authors · 2023-08-28
>
> We would like to sincerely thank you for your careful consideration of our paper and for providing valuable feedback. Please see below for our responses.
>
> R1. Number of runs and variance of LLMs: We chose to do 3 runs to balance between efficiency and effectivness. We acknowledge that due to the inherent variance in LLMs, a larger number of runs might provide a more comprehensive understanding of the model's behavior. We will conduct additional experiments with a greater number of runs in a revision.
>
> R2. Context details and data sources: We apologize for any ambiguity in our presentation. The arguments used as <context> in our prompt are from training set of IBM30k. Since AQ is a universal concept, the AQA effectiveness is evaluated on a mixture of arguments from different topics in prior art [1,2]. Following this paradigm, when selecting arguments to serve as examples, we chose arguments from different topics and did not impose specific requirements on stance of the arguments. The chosen examples are of diverse scores to enable them to represent arguments of varying quality. We hope that by providing LLMs with examples of arguments from different topics and stances, we can enhance its robustness in scoring arguments. We attempted to use arguments from the same topic as examples in the prompts. As prelimenary evaluation showed that this had little impact on the results, we did not investigate further with this setting.
>
> R3. Number of arguments as examples: We appreciate this concern and understand the potential impact of example size on the model's performance. To address this, we will experiment with varying context sizes to determine whether there is an optimal threshold beyond which further improvement in scoring becomes marginal.
>
> QA: See reply to R2.
>
> QB: A good argument should stay on topic during the process of reasoning and maintain strong relevance to the subject. Therefore, in our input, we have concatenated the topic with the argument. In this way, the argument representation has taken the related topic into consideration, which is further updated through interaction with other arguments.
>
> [1] Shai Gretz, Roni Friedman, Edo Cohen-Karlik, Assaf Toledo, Dan Lahav, Ranit Aharonov, Noam Slonim: A Large-Scale Dataset for Argument Quality Ranking: Construction and Analysis. AAAI 2020: 7805-7813
>
> [2] Favreau C O, Zouaq A, Bhatnagar S. Learning to Rank with BERT for Argument Quality Evaluation[C]//The International FLAIRS Conference Proceedings. 2022, 35.

---

### Meta-Review · Area_Chair_fH8R · 2023-09-19

**Recommendation:** 5

**Metareview:**

The reviewers agreed on the importance of analyzing and understanding the structural similarity between arguments to assess their quality. They found the approach novel and sound. The reviewers also appreciated the author rebuttal which clarified the issues highlighted in the reviews.

---

### Decision · Program_Chairs · 2023-10-07

**Decision:**

Accept-Main

**Comment:**

The reviewers agreed on the importance of analyzing and understanding the structural similarity between arguments to assess their quality. They found the approach novel and sound. The reviewers also appreciated the author rebuttal which clarified the issues highlighted in the reviews.